# Smad7 Deficiency in Myeloid Cells Does Not Affect Liver Injury, Inflammation or Fibrosis after Chronic CCl_4_ Exposure in Mice

**DOI:** 10.3390/ijms222111575

**Published:** 2021-10-27

**Authors:** Ludmilla Unrau, Jessica Endig, Diane Goltz, Paulina Sprezyna, Hanna Ulrich, Julia Hagenstein, Bernd Geers, Karina Kaftan, Lukas Carl Heukamp, Gisa Tiegs, Linda Diehl

**Affiliations:** 1Institute of Experimental Immunology and Hepatology, University Medical Center Hamburg-Eppendorf, Martinistrasse 52, 20246 Hamburg, Germany; ludmilla.unrau@bnitm.de (L.U.); endig.jessica@gmail.com (J.E.); Sprezyna@gmx.de (P.S.); h.ulrich@uke.de (H.U.); j.hagenstein@uke.de (J.H.); bernd.geers@gmx.de (B.G.); Karina.Kaftan@haw-hamburg.de (K.K.); g.tiegs@uke.de (G.T.); 2Hamburg Center for Translational Immunology (HCTI), University Medical Center Hamburg-Eppendorf, Martinistrasse 52, 20246 Hamburg, Germany; 3Department of Virology, Bernhard Nocht Institute for Tropical Medicine, Bernhard-Nocht-Str. 74, 20359 Hamburg, Germany; 4Institute for Hematopathology, Fangdieckstrasse 75a, 22547 Hamburg, Germany; goltz@hp-hamburg.de (D.G.); heukamp@hp-hamburg.de (L.C.H.)

**Keywords:** chronic liver injury, CCl_4_, Smad7, TGF-β, myeloid cell, inflammation, fibrosis

## Abstract

Myeloid cells play an essential role in the maintenance of liver homeostasis, as well as the initiation and termination of innate and adaptive immune responses. In chronic hepatic inflammation, the production of transforming growth factor beta (TGF-β) is pivotal for scarring and fibrosis induction and progression. TGF-β signalling is tightly regulated via the Smad protein family. Smad7 acts as an inhibitor of the TGF-β-signalling pathway, rendering cells that express high levels of it resistant to TGF-β-dependent signal transduction. In hepatocytes, the absence of Smad7 promotes liver fibrosis. Here, we examine whether Smad7 expression in myeloid cells affects the extent of liver inflammation, injury and fibrosis induction during chronic liver inflammation. Using the well-established model of chronic carbon tetrachloride (CCl_4_)-mediated liver injury, we investigated the role of Smad7 in myeloid cells in LysM-Cre Smad^fl/fl^ mice that harbour a myeloid-specific knock-down of Smad7. We found that the chronic application of CCl_4_ induces severe liver injury, with elevated serum alanine transaminase (ALT)/aspartate transaminase (AST) levels, centrilobular and periportal necrosis and immune-cell infiltration. However, the myeloid-specific knock-down of Smad7 did not influence these and other parameters in the CCl_4_-treated animals. In summary, our results suggest that, during long-term application of CCl_4_, Smad7 expression in myeloid cells and its potential effects on the TGF-β-signalling pathway are dispensable for regulating the extent of chronic liver injury and inflammation.

## 1. Introduction

Transforming growth factor β (TGF-β)-dependent signalling is consistently detected in fibrotic tissues. Diverse causes, ranging from toll-like receptor (TLR)-signalling oxidative stress to proinflammatory cytokines, can provoke transcription of TGF-β in fibrotic tissues [1,2]. Immune cells can influence fibrosis development, either by producing cytokines and growth factors or via the release of proteases capable of extra cellular matrix (ECM) modulation [3]. As such, macrophages can be both producers and responders to TGF-β. For instance, disruption of macrophage recruitment inhibits fibrosis development, which is associated with reduced TGF-β synthesis [1]. Moreover, prevention of TGF-β signalling in macrophages, by use of a macrophage-specific, TGF-β receptor type I (TβRI) [4] or a Smad3-deficient mouse model [5], reduces tissue fibrosis, which may be due to attenuated oxidative burst capacity and a reduction in proinflammatory gene transcription [6,7]. Additionally, the attenuation of TGF-β signalling in macrophages can lead to reduced expression of pro-fibrotic cytokines, thereby lessening tissue fibroblast activation and tissue fibrosis indirectly [7,8].

TGF-β signalling via TβRI leads to phosphorylation of the receptor-activated Smads (R-Smads) Smad2 and 3. These form complexes with Smad4 that translocate to the nucleus, where they regulate gene transcription. TGF-β signalling is tightly regulated at different levels [1,9]. For one, TGF-β signalling itself induces the expression of inhibitory Smads (Smad6 and Smad7), which in turn antagonise the TGF-β-signalling pathway in different ways, which include interfering with TGF-β receptor kinase activity, interrupting complex formation between R-Smads and Smad4 and promoting activity of E3 ubiquitin ligases, leading to enhanced degradation of R-Smads [9] or TβRI [10]. Thus, the induction and overexpression of Smad7 was found to consistently attenuate fibrosis in many pathophysiologically distinct fibrotic conditions. On the contrary, increased expression of Smad7 in the intestine was associated with enterocolitis [11] and refractory coeliac disease [12], which was associated with overexpression of proinflammatory cytokines a.o. by macrophages [12].

In the context of chronic liver injury and inflammation, TGF-β signalling in hepatic stellate cells is well known to promote liver fibrosis [13], and complete deletion of the TGF-β-signalling attenuator Smad7 [14], or specifically in hepatocytes [15], leads to enhanced fibrosis and accelerated hepatocellular carcinoma (HCC) development, respectively. Although macrophages can both produce and respond to TGF-β, it is unclear, however, how the negative regulator of TGF-β-signalling Smad7 in macrophages/myeloid cells impacts hepatic injury and inflammation. Here, we investigated the effect of myeloid-specific Smad7 deficiency with the use of a floxed Smad7 mouse line crossed with a LysM-Cre mouse line, in chronic liver injury, inflammation and fibrosis induced by chronic CCl_4_ application.

## 2. Results

### 2.1. Myeloid-Specific Knock-Down of Smad7 Does Not Affect Liver Injury after Chronic CCl_4_ Exposure

Using a myeloid-specific, Cre-mediated, Smad7 knock-down mouse model, in which efficient knock-down of *Smad7* mRNA in myeloid cells, including splenic CD11b^pos^ cells (Endig et al.) [16] and CD64^pos^Clec2a^pos^ hepatic Kupffer cells (Appendix A) was present, we investigated the extent of liver injury, inflammatory cell infiltration and fibrosis development after chronic CCl_4_ exposure. To this end, LysM-Cre^pos^ Smad7^Δ^^/^^Δ^ and Smad7^fl/fl^ littermate controls were treated with corn oil as a control or with CCl_4_ in corn oil for 6 weeks. Treatment with CCl_4_ led to highly increased serum ALT and AST levels that were not observed in corn-oil-treated controls (Figure 1A). However, the absence of Smad7 in myeloid cells did not change the extent of liver injury as measured by ALT/AST. CCl_4_ treatment also consistently reduced the body weight of mice, but neither liver weight, body weight nor body-to-liver weight ratio was affected by myeloid Smad7 deficiency (Figure 1B). Histological haematoxylin and eosin (H&E) and Masson–Goldner (Figure 1C) staining revealed highly increased levels of infiltrating immune cells (neutrophils, eosinophils), fibrosis and necrosis (Figure 1C,D). However, here, the presence or absence of Smad7 expression did not alter the extent of these inflammatory changes and fibrosis markers, indicating that the TGF-β negative regulator Smad7, when absent in myeloid cells, did not influence the extent of immune cell infiltration or the degree of fibrosis after chronic CCl_4_ exposure.

### 2.2. Comparable Myeloid-Cell Infiltration in Livers of Myeloid-Cell-Specific, Smad7-Deficient Mice and Wild-Type Littermates after Chronic CCl_4_ Treatment

Although the histological analysis did not reveal large differences in hepatic immune-cell infiltration after CCl_4_ treatment between animals lacking myeloid Smad7 expression and their wild-type littermate controls, we proceeded to analyse in more detail whether the absence of Smad7 expression in myeloid cells altered the composition of the infiltrating myeloid-cell compartment in the livers of CCl_4_-treated animals. We focused on the infiltration of total CD11b^pos^ myeloid cells and the presence of subpopulations of neutrophilic granulocytes (CD11b^pos^CD11c^neg^Ly6G^pos^), eosinophilic granulocytes (CD11b^pos^CD11c^neg^SiglecF^pos^) and inflammatory monocytes (CD11b^pos^CD11c^neg^Ly6G^neg^ SiglecF^neg^Ly6C^high^) (Figure 2A).

Here, we observed significant increases in CD11b^pos^ cells after CCl_4_ treatment (Figure 2B), which, however, did not differ between Smad7-proficient and -deficient animals. In corn-oil-treated animals, Smad7 deficiency led to a reduction in the percentage of Ly6G^pos^ cells within CD11b^pos^ cells. However, as we have no data from untreated LysM-Cre Smad7^fl/fl^ animals, we cannot define whether this was oil-induced or a steady-state alteration. Within the CD11b^pos^ compartment in CCl_4_-treated mice, neutrophils, eosinophils and inflammatory monocyte percentages were similar between Smad7-proficient and -deficient animals (Figure 2B). mRNA-expression analysis revealed a significant increase in fibrosis-associated αSMA (*acta2*) expression and a tendency for increased mRNA expression of TNFα (*Tnf*), IL-10 (*Il10*) and CCL2 (*Ccl2*) in CCl_4_-treated animals, which was not altered in the absence of myeloid-expressed Smad7 (Figure 2C). *Mertk* mRNA levels did not seem to increase after CCl_4_ treatment. To determine whether the absence of Smad7 in myeloid cells influenced overall TGF-β signalling in the liver, we analysed whole-liver *Tgfb*, *Tgfbr1* and *Smad3* mRNA expression by quantitative RT-PCR (qPCR) (Figure 2D). Interestingly, hepatic *Smad3* mRNA was significantly increased in corn-oil-treated mice in the absence of myeloid Smad7. Furthermore, we detected a trend for increased *Tgfb*, but not *Tgfbr1* and *Smad3* mRNA expression in CCl_4_-treated animals, which did not depend on Smad7 expression in myeloid cells (Figure 2D). In summary, these data indicated that after CCl_4_-induced liver injury, the absence of Smad7 in LysM-expressing myeloid cells did not change the composition of hepatic infiltrating cells or the mRNA expression profile.

### 2.3. Similar Cytokine Production by Intrahepatic Non-Parenchymal Cells (NPCs) in the Presence and Absence of Myeloid-Expressed Smad7 after Chronic CCl_4_-Mediated Liver Injury

As our mRNA analyses were performed on whole-liver tissue, which may not wholly represent protein production, we decided to investigate the cytokine-producing capability of liver-infiltrating immune cells by a flow-cytometry-based multiplex assay. Liver NPCs were isolated by collagenase digestion and density-gradient centrifugation and stimulated with PMA, ionomycin and LPS for 24 h, after which supernatants were tested for the presence of various pro- and anti-inflammatory cytokines. We detected increased production of IFNγ, TNFα, IL-17, CCL2 and IL-10 due to exposure to CCl_4_ by hepatic NPCs, but found no evidence that myeloid-expressed Smad7 influenced the expression of these cytokines by NPCs (Figure 3). Although in this setting, we did not determine exactly which cells produced the detected cytokines, we concluded that both direct and indirect functions of myeloid cells, for instance by influencing T-cell cytokine production, were not affected by the absence of Smad7 expression.

### 2.4. The Lack of Smad7 in Myeloid Cells Does Not Affect Liver Regeneration

As TGFβ signalling promotes fibrosis and wound healing, we examined whether the lack of the negative regulator Smad7 in myeloid cells influenced the regenerative response of the liver during chronic CCl_4_-mediated liver injury. To this end, we analysed the mRNA expression of several cell cycle-associated genes that are well known in combination with mediated hepatocyte proliferation after liver damage (Figure 4). A trend for increased expression of *Ccnb1*, *Ccnd1* and *Cdk4* mRNA in response to CCl_4_ treatment was observed, which did not differ due to myeloid Smad7 deficiency. After CCl_4_ treatment, a significant increase in hepatic *Cdk1*, *Cdc25a* and *Cdkn1a* mRNA was observed, which again did not change between LysM-Cre^pos^ Smad7^Δ^^/^^Δ^ and Smad7^fl/fl^ groups. Additionally, we probed CCl_4_ livers for the expression of the proliferation marker Ki-67, both by histology (Figure 4B) and qPCR (Figure 4C). We observed an increased presence of Ki-67 positive hepatocytes in CCl_4_-treated animals (Figure 4B). This was mirrored by the increased presence of *Ki-67* mRNA in CCl_4_-treated mice (Figure 4C). However, the regenerative response of livers after CCl_4_ treatment was not altered by the absence of Smad7 in myeloid cells.

## 3. Discussion

TGF-β production and signalling play an important role in fibrogenic responses in many organs, including the liver [17,18]. TGF-β signalling can be regulated by the induction of inhibitory Smad6 and 7, which render cells unable to properly detect TGF-β signals via the down-regulation of the TβRI or R-Smads [9,10]. It has previously been described that the prevention of TGF-β signalling in macrophages, by use of a macrophage-specific TβRII [4], reduces tissue fibrosis, possibly by affecting oxidative burst capacity and/or proinflammatory gene transcription [6,7]. Additionally, the attenuation of TGF-β signalling in macrophages reduces their expression of pro-fibrotic cytokines, which could indirectly lead to less tissue fibroblast activation and thus tissue fibrosis [7,8].

High Smad7 expression is associated with several inflammatory conditions, including CMV colitis [19] and necrotizing enterocolitis [11], but, when overexpressed, can also attenuate liver fibrosis [20]. These seemingly contradictory results may be due to differential effects that Smad7 overexpression or deletion may have on different cell types. For one, the overexpression of Smad7 could impede TGF-β-mediated down-regulation of proinflammatory cytokine synthesis [11,12,19], leading to the enhancement of inflammatory conditions. On the contrary, at the same time, it may also inhibit the profibrogenic action of TGF-β, thereby leading to less pronounced fibrosis development [14,20].

Here, we analysed whether Smad7 function in LysM-expressing myeloid cells influenced the induction of liver fibrosis in the model of chronic CCl_4_ application. We previously established a myeloid-specific knock-down mouse line by crossing a LysM-Cre expressing line with a floxed Smad7 line, in which we confirmed the lack of Smad7 expression in splenic CD11b^pos^ myeloid cells [16] and hepatic Kupffer cells (Appendix A). After extensive analysis of various parameters in chronically CCl_4_-treated LysM-Cre^pos^ x Smad7^Δ^^/^^Δ^ mice and their LysM-Cre^neg^ Smad7^fl/fl^ littermate controls, our results indicated that liver fibrosis development did not depend on the presence of Smad7 in LysM-expressing cells. Although CCl_4_-treatment led to increased serum ALT and AST levels and increased centrilobular and periportal necrosis, the lack of Smad7 in myeloid cells did not affect the levels of liver enzymes and liver damage (Figure 1). Liver-infiltrating myeloid-cell populations (Figure 2) were also not affected by myeloid-specific Smad7 ablation in CCl_4_-treated animals. Interestingly, in several conditions of gut inflammation, the induction of Smad7 in macrophages is associated with higher inflammatory cytokine production [11,19]. Here, we detected neither a decrease nor an increase in hepatic proinflammatory cytokine RNA expression (Figure 2) in CCl_4_-treated animals as a consequence of myeloid-specific Smad7 deletion. As cytokine mRNA levels in whole-liver samples may not be specific enough to detect differences in cytokine production by a small population of liver NPCs, we analysed cytokine production of liver NPCs ex vivo (Figure 3) at the protein level, but again did not find evidence for Smad7-regulated myeloid-cell function in NPCs from chronically inflamed livers. A possible explanation for these results could stem from the difference between the effects reported in the literature after overexpression of Smad7, which leads to the inhibition of the anti-inflammatory effects of TGF-β signalling, and thus conversely to enhanced inflammatory cytokine production [11,19]. In the absence of Smad7 in myeloid cells, the capacity of these cells to circumvent the anti-inflammatory effects of TGF-β signalling may be limited and therefore may have no large effect on cytokine production. Additionally, Smad7 activity in other hepatic cell populations, such as hepatocytes and stellate cells/myofibroblasts, may have a larger impact on liver fibrosis, when lacking Smad7, as it is reported that global and hepatocyte-specific, Smad7-deficient mice have pronounced aberrant phenotypes in diverse liver disease models, such as CCl_4_- and alcohol-induced liver injury and hepatocellular carcinoma [14,15,20,21]. Smad7 deficiency did seem to influence neutrophil infiltration and Smad3 mRNA expression in corn-oil-treated animals. These effects were unexpected and, at this time, prove difficult to interpret as we lack data concerning these parameters in non-treated animals. Thus, we cannot definitely discern between these effects being caused by corn-oil treatment or being in place under homeostatic conditions.

Finally, we examined the effect of myeloid-expressed Smad7 on the capacity of the liver to regenerate during chronic CCl_4_ application. The liver is known for its regeneration ability via the induction of hepatocyte proliferation and differentiation [22], a process that relies heavily on various signals from cytokines. In this context, TGF-β signalling is known to inhibit regeneration by direct action on hepatocytes [22,23]. Additionally, resolution of injury and fibrosis development in the liver also depends on macrophage phenotype and function [24,25] and other cells from the myeloid lineage, like neutrophils [26] and eosinophils [27,28]. Our analyses revealed that, although there was a general induction of cell cycle genes during CCl_4_ treatment, there were no clear general changes due to myeloid Smad7 deficiency. Overall, the comparable expression of *Ki67* mRNA between LysM-Cre^pos^ Smad7^Δ^^/^^Δ^ and Smad7^fl/fl^ littermates after CCl_4_ application (Figure 4) showed that the number of proliferating hepatocytes did not change due to myeloid-specific Smad7 deficiency.

In conclusion, our data suggest that the negative regulator of TGF-β-signalling Smad7, when expressed in myeloid cells, has no influence on the severity of liver damage, the infiltrating myeloid-cell compartment, the cytokine profile of these infiltrating cells and the regenerative response in chronic CCl_4_-induced liver injury, inflammation and fibrosis.

## 4. Materials and Methods

### 4.1. Mice

B6.129P2-Lyz2^tm1(cre)Ifo^/J and B6.Cg-Smad7^tm1.1Ink^/J (LysM-Cre Smad7^fl/fl^) mice were back-crossed and bred under specific pathogen-free conditions in the central animal facility of the University Medical Center Hamburg-Eppendorf according to the Federation of Laboratory Animal Science Association guidelines. For the experiments, 8-to-12 week-old males were used. The Behörde für Soziales, Familie, Gesundheit und Verbraucherschutz (Hamburg, Germany; approval code 78/16) approved all experiments. All mice were kept under ad libitum supply of food and water and a 12/12 h day–night rhythm. All efforts were made to minimise suffering.

### 4.2. Induction and Evaluation of Chronic Liver Inflammation and Fibrosis Using Carbon Tetra Chloride

CCl_4_ (Sigma-Aldrich, Darmstadt, Germany) was diluted in corn oil (Sigma-Aldrich) to generate a 30% (*w*/*v*) solution of CCl_4_. Mice were injected twice per week with 0.6 mL/kg of CCl_4_ in corn oil or as a control, received only corn oil for the indicated treatment duration. Mice were sacrificed by an intravenous injection of a ketamine–xylazine–heparin solution. Serum levels of ALT (20764957322) and AST (20764949322) were measured using a Cobas Mira Chemistry Analyser (Roche, Mannheim, Germany). Liver tissue was isolated for RNA, cDNA synthesis and qPCR to quantify intrahepatic levels of cytokine mRNA or cell cycle gene mRNA. Histology of liver tissue was performed on formalin-fixed paraffin-embedded liver samples.

### 4.3. Isolation and Stimulation of Non-Parenchymal Liver Cells

Non-parenchymal liver cells (NPCs) were isolated according to standard protocols. In brief, livers were mechanically disrupted and digested with collagenase IV (Sigma-Aldrich), after which NPCs were enriched via a 40–80% Percoll gradient (GE Healthcare, Solingen, Germany). For quantification of cytokine production by NPCs, 10^5^ cells were plated into a 96-well U-bottom plate in RMPI medium containing 8% FCS, pen/strep, glutamine and β-mercaptoethanol in the presence or absence of PMA (5 ng/mL), ionomycin (200 ng/mL) (both from Sigma-Aldrich) and LPS (1 μg/mL) (Invivogen, Toulouse, France). After 24 h, supernatants were collected and stored at −20 °C until further analysis. Proinflammatory cytokine production was measured via the LEGENDplex (BioLegend, San Diego, California, USA) Mouse Inflammation Panel (IL-23, IL-1α, IFNγ, TNFα, MCP-1, IL-12p70, IL-1β, IL-10, IL-6, IL-27, IL17A, IFNβ, GM-CSF), according to the manufacturer’s instructions. The cytokines IL-27, IL-12p70, IFNβ and GM-CSF were all below detection levels.

### 4.4. Flow Cytometry

Flow cytometric analyses were conducted on a FACSCanto II or LSR II (BD Biosciences, Franklin Lakes, NJ, USA). All antibodies were from BioLegend or eBioscience (Thermo Fisher Scientific, Bremen, Germany), unless otherwise stated. LIVE/DEAD Fixable Violet or Near-IR Dead Cell Stain Kit (Invitrogen, Thermo Fisher Scientific) was used to exclude dead cells in all samples analysed. Anti-CD16/32 antibody (clone 2.4G2) was included in each staining at 10 μg/mL to block unspecific antibody binding via Fc receptors. Leukocytes were stained with antibodies against murine CD11b (clone M1/70), CD11c (clone N418), SiglecF (clone E50-2440), Ly6G (clone 1A8), Ly6C (clone HK1.4), CD64 (clone X54-5/7.1) and Clec2a (clone 17D9). Data were analysed using FlowJo software version 10 (Becton, Dickinson & Company, Ashland, OR, USA).

### 4.5. mRNA Isolation and Quantitative RT-PCR

Murine tissue or cells were flash frozen in liquid nitrogen and stored at −80 °C until further processing. mRNA was isolated using a RNeasy Mini Kit (Qiagen, Hilden, Germany), following the manufacturer’s instructions. For tissue probes, genomic DNA was digested separately using the DNase Treatment and Removal Kit (AM1906, Invitrogen, ThermoFisher Scientific). mRNA was transcribed into cDNA using the High-Capacity cDNA Reverse Transcription Kit (Applied Biosystems, Thermo Fisher Scientific). Quantitative RT-PCR (qPCR) was performed with either exon-spanning primers for relevant inflammatory and cell cycle genes using the PowerUp SYBR Green Master Mix (A25742, Applied Biosystems) or with the TaqMan Gene Expression Assays (Applied Biosystems) for Ki-67 (Mm01278617_m1) and the reference genes muGAPDH (Mm99999915_g1)/muHprt1 (Mm00446968_m1) on a ViiA QuantStudio 7 (Thermo Fischer Scientific). Primers (Metabion, Planegg, Germany) used for SYBR Green qPCR were: muIl1b forward: 5′-TGCCACCTTTTGACAGTGATGA-3′, muIl1b reverse: 5′-TGATGTGCTGCTGCGAGATT-3′, muCcl2 forward: 5′-TGGAGCATCCACGTGTTGG-3′, muCcl2 reverse: 5′-ACCTCTCTCTTGAGCTTGGTG-3′, muActa2 forward: 5′- CCAGCCATCTTTCATTGGGATG-3′, muActa2 reverse: 5′- TATAGGTGGTTTCGTGGATGCC-3′, muTnf forward: 5′-CCACCACGCTCTTCTGTCTAC-3′, muTnf reverse: 5′-AGGGAGGCCATTTGGGAACT-3′, muIl10 forward: 5′-ATAAGAGCAAGGCAGTGGAGC-3′, muIl10 reverse: 5′-AAGATGTCAAATTCATTCATGGCCT-3′, muMertk forward: 5′-GGCTCTGCAAGGTAAGCTCG-3′, muMertk reverse: 5′-AAATGCTGGGTCCGAAGCTC-3′, muCcnd1 forward: 5′-TCAAGTGTGACCCGGACTG-3′, muCcnd1 reverse: 5′-CAGCCTCTTCCTCCACTTCC-3′, muCdk4 forward: 5′-GAGCGTAAGATCCCCTGCTT-3′, muCdk4 reverse: 5′-ACCGACACCAATTTCAGCCA-3′, muCdk2 forward: 5′-CATCTTTGCTGAAATGCACCTAGT-3′, muCdk2 reverse: 5′-TCCTTGTGATGCAGCCACTT-3′, muCdc25a forward: 5′-GCGGTGTTGAAGAGAGCAGA-3′, muCdc25a reverse: 5′-GAGACTGGGATGGAAGCTGG-3′, muCdkn1a forward: 5′-AGTGTGCCGTTGTCTCTTCG-3′, muCdkn1a reverse: 5′-AAGTTCCACCGTTCTCGGG-3′, muCdk1 forward: 5′-CACACGAGGTAGTGACGCTG-3′, muCdk1 reverse: 5′-TCTGAGTCGCCGTGGAAAAG-3′, muCcnb1 forward: 5′-CAACGGTGAATGGACACCAA-3′, muCcnb1 reverse: 5′-TATGTACAGGCGGCACATGG-3′, mu18S forward: 5′-CACGGCCGGTACAGTGAAAC-3′, mu18S reverse: 5′-AGAGGAGCGAGCGACCAA A-3′, muHprt1 forward: 5′-TGCTGACCTGCTGGATTACATT-3′, muHprt1 reverse: 5′-CTTTTATGTCCCCCGTTGACTG-3′, muEef1g forward: 5′-GTCTGTACCCTGTTGTGGCT-3′, muEef1g reverse: 5′-TCCCCCAAGATAGCCCTGAA-3′. muTgfb1 forward: 5′-AGCGGACTACTATGCTAAAGAGG-3′, muTgfb1 reverse: 5′- ATGGGGGTTCGGGCACT-3′: muTgfbr1 forward: 5′-CGACGCTGTTCTATTGGTGG-3′, muTgfbr1 reverse: 5′-TCACTCTCAAGGCCTCACAG-3′, muSmad3 forward: 5′-AAGAAGCTCAAGAAGACGGGG-3′, muSmad3 reverse: 5′-ACAGGCGGCAGTAGATAACG-3′. Relative mRNA expression levels were calculated with the ΔCt method.

### 4.6. Histology

Liver tissue was fixed in 4% (*w*/*v*) formalin (J.T. Baker, Fisher Scientific, Schwerte, Germany) and processed in routine paraffin embedding. Haematoxylin and eosin (H&E) and Masson–Goldner staining was performed using standard protocols. Liver sections were stained with anti-Ki-67 (Roche MIB-1 RTU) on a LeicaBond system using a Bond Intense R kit (Leica Biosystems, Wetzlar, Germany). 

### 4.7. Statistical Analysis

Student’s *t*-test or one-way analysis of variance (ANOVA) test with Tukey’s multiple comparisons correction test was used to determine statistical significance of the results. In data sets where the Brown–Forsythe analysis suggests that data are not distributed in a Gaussian manner, data were alternatively analysed via a non-parametric Kruskal–Wallis test applying a Dunn’s multiple comparisons analysis. Data are depicted as the mean +/− SEM, and *p*-values < 0.05 were considered significant.

## Figures and Tables

**Figure 1 ijms-22-11575-f001:**
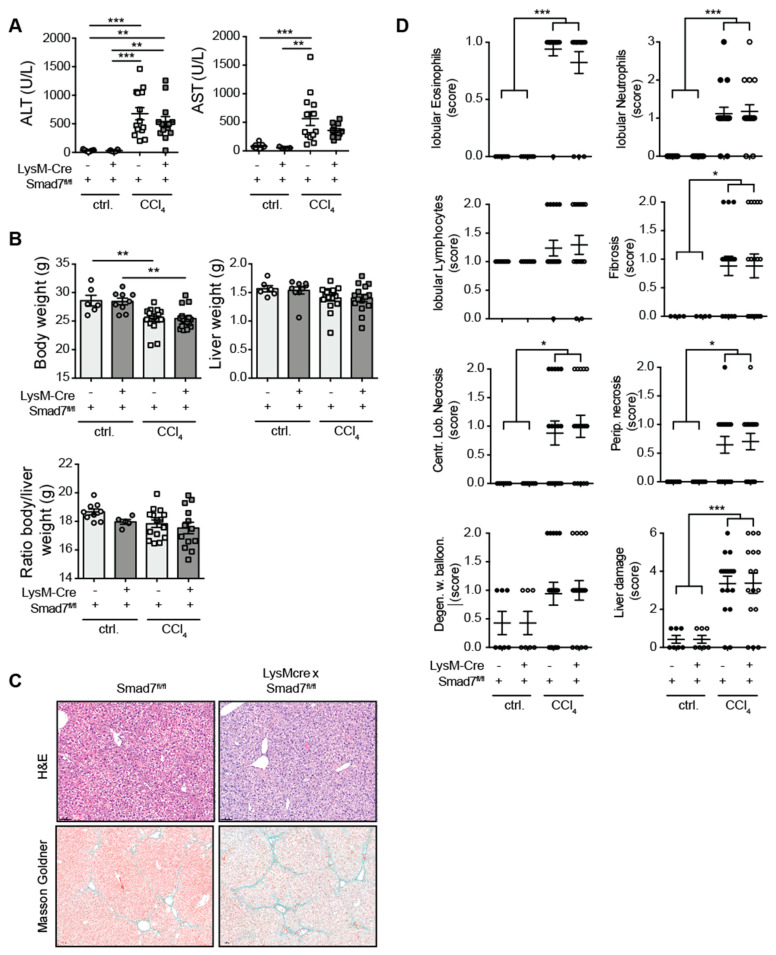
Liver enzymes and histological damage parameters were not changed due to myeloid deletion of Smad7 after chronic CCl_4_ application. LysM-Cre^pos^ Smad7^Δ^^/^^Δ^ and Smad7^fl/fl^ littermates were injected intraperitoneally with CCl_4_ in corn oil or corn oil alone (ctrl.) twice a week for 6 wks. Mice were sacrificed 24 h after the last CCl_4_–oil application. (**A**) Serum ALT and AST (U/L). (**B**) Body weight, liver weight and body-to-liver weight ratio of mice at the time of sacrifice. (**C**) Representative H&E and Masson–Goldner staining of livers of CCl_4_-treated Smad7^fl/fl^ and LysM-Cre^pos^ Smad7^Δ^^/^^Δ^ littermates at 10× magnification as indicated. (**D**) Histological scores of H&E staining in which the liver damage score is a cumulative value out of periportal (Perip.) and centrilobular (Centr. Lob.) necrosis, degeneration with ballooning (Degen. w. balloon.) and fibrosis. Data shown are cumulative data of 5 independent experiments with 3–5 animals per group. Data are displayed as the mean ± SEM. Data were analysed by ANOVA: * *p* ≤ 0.05, ** *p* ≤ 0.01, *** *p* ≤ 0.001.

**Figure 2 ijms-22-11575-f002:**
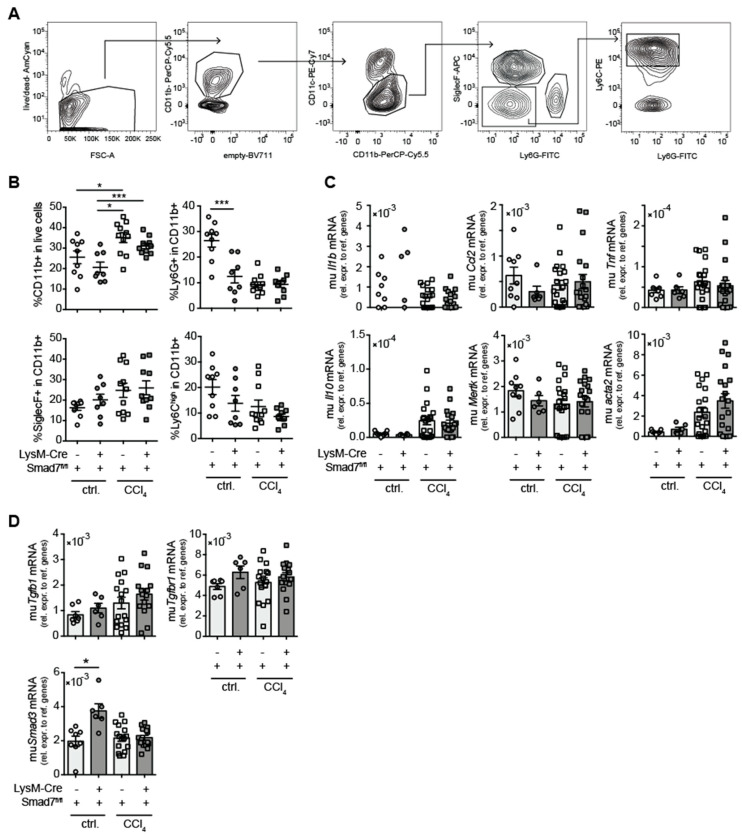
Myeloid deletion of Smad7 changed neither hepatic myeloid-cell infiltration nor hepatic expression of (anti-) inflammatory and fibrosis markers after chronic CCl_4_ application. LysM-Cre^pos^ Smad7^Δ^^/^^Δ^ and Smad7^fl/fl^ littermates were injected intraperitoneally with CCl_4_ in corn oil or corn oil alone (ctrl.) twice a week for 6 wks. Mice were sacrificed 24 h after the last CCl_4_–oil application. (**A**) Gating strategy of flow cytometric analysis of myeloid cells after gating on single viable lymphocytes in hepatic, non-parenchymal cells (NPCs) from the liver: neutrophilic granulocytes were defined as CD11b^pos^CD11c^neg^Ly6G^pos^SiglecF^neg^, eosinophils as CD11b^pos^CD11c^neg^ Ly6G^neg^SiglecF^pos^ and inflammatory monocytes as CD11b^pos^CD11c^neg^Ly6G^neg^SiglecF^neg^ Ly6G^high^. (**B**) Percentages of different myeloid cells as defined in A. (**C**) Pro- (*Il1b*, *Ccl2*; *Tnf*) and anti- (*Il10*) inflammatory cytokine and resolution/fibrosis (*Mertk*, *Acta2*) marker gene expression in the livers of LysM-Cre^pos^ Smad7^Δ^^/^^Δ^ and Smad7^fl/fl^ littermates measured by qPCR. (**D**) mRNA expression of TGF-β-signalling relevant molecules *Tgfb1*, *Tgfbr1* and *Smad3*. Data shown are cumulative data of 5 independent experiments with 3–5 animals per group. Data are displayed as the mean ± SEM. Data were analysed by ANOVA: * *p* ≤ 0.05, *** *p* ≤ 0.001.

**Figure 3 ijms-22-11575-f003:**
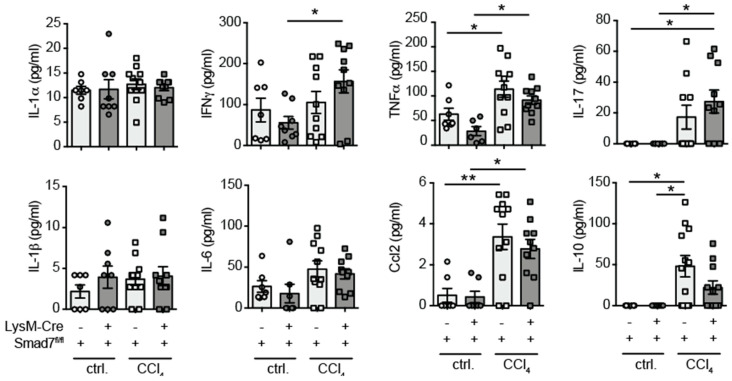
Proinflammatory cytokine production by hepatic NPCs was not influenced by the absence of Smad7. LysM-Cre^pos^ Smad7^Δ^^/^^Δ^ and Smad7^fl/fl^ littermates were injected intraperitoneally with CCl_4_ in corn oil or corn oil alone (ctrl.) twice a week for 6 wks. Mice were sacrificed 24 h after the last CCl_4_–oil application. NPCs isolated from the liver were stimulated in vitro with PMA, ionomycin and LPS. Supernatants were collected 24 h later, and cytokine content was measured using the LEGENDplex 13× Mouse Inflammation Panel. Data shown are cumulative data of 5 independent experiments with 3–5 animals per group. Data are displayed as the mean ± SEM. ANOVA analysis of the data did not reveal significant differences between LysM-Cre^pos^ Smad7^Δ^^/^^Δ^ and Smad7^fl/fl^ groups after CCl_4_ treatment. * *p* ≤ 0.05, ** *p* ≤ 0.01.

**Figure 4 ijms-22-11575-f004:**
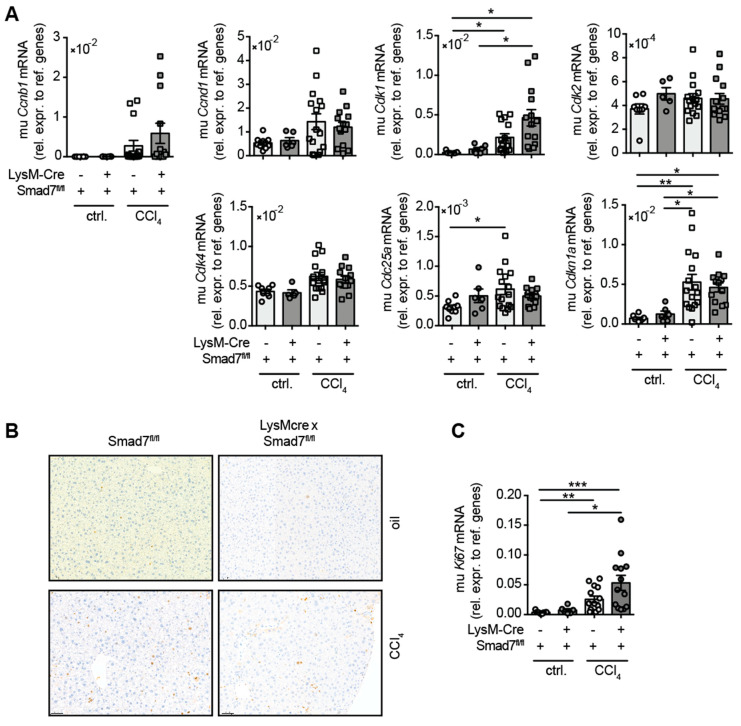
Hepatic expression of cell cycle genes in CCl_4_-treated Smad7-proficient and -deficient animals. LysM-Cre^pos^ Smad7^Δ^^/^^Δ^ and Smad7^fl/fl^ littermates were injected intraperitoneally with CCl_4_ in corn oil or corn oil alone (ctrl.) twice a week for 6 wks. Mice were sacrificed 24 h after the last CCl_4_–oil application. (**A**) Cell cycle gene expression in the livers of LysM-Cre^pos^ Smad7^Δ^^/^^Δ^ and Smad7^fl/fl^ littermates measured by qPCR. (**B**) Histological staining of Ki-67 in liver sections of CCl_4_ or corn-oil-treated mice shown at 10x magnification. (**C**) Expression of *Ki67* mRNA in livers of treated mice as described above. Data shown are cumulative data of 5 independent experiments with 3–5 animals per group. Data are displayed as the mean ± SEM. ANOVA analysis of the data did not reveal significant differences between LysM-Cre^pos^ Smad7^Δ^^/^^Δ^ and Smad7^fl/fl^ groups after CCl_4_ treatment. * *p* ≤ 0.05, ** *p* ≤ 0.01, *** *p* ≤ 0.001.

## Data Availability

The data presented in this study are available on request from the corresponding author.

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
