# Peer review of "Smad7 Deficiency in Myeloid Cells Does Not Affect Liver Injury, Inflammation or Fibrosis after Chronic CCl4 Exposure in Mice"

_ijms, 2021, doi:10.3390/ijms222111575_

Round 1
Reviewer 1 Report
This paper aims to present the involvement of the SMAD7 in myeloid cells in liver injury, inflammation and fibrosis employing the well-established animal model of liver injury (CCL4).
The work has been well designed, the methodology employed is adequate, and the results are properly analysed and described. The only “but” that could be mentioned to the work is the lack of positive results. The authors have thoroughly analysed parameters related with liver damage, and tried to confirm the involvement of the SMAD7 deletion in the selected model. Unfortunately, all the biochemical changes described all along the paper are those related to the expose to the toxic (CCL4) and none to the deletion of the SMAD7.
The paper is easy to read, and is for the interest of researchers in the field of liver disease and TGF-β/SAMD7 signalling.
General comments:
- It could be of interest to present (at least once as “supplementary material”) some experimental evidence of the lack of myeloid expression of Samd7 in the samples employed.
- There is only one “positive” effect of the deletion of the SAMD7, that is the increase in the mRNA expression of Cdk1 (figure 4A). I suggest authors to confirm this increase at least at protein level (Western blot), explore the molecular mechanism involved, and discuss the consequences.
- In this study, authors have employed only mice males. Researches should start working with male and female animals to detect possible sex-induced differences.
Minor points:
- Page 2 line 57. There should be a reference for the previous statement. “The induction and overexpression of Smad7 thus has been found to consistently attenuate fibrosis in many patho-physiologically distinct fibrotic conditions”
- The first line of the footnotes in Fig 1 and Fig 2 should be in bold.
- Page 4 line 110: There should be an space between (anti)inflammatory….
- The source of several reagents should be included in the material and methods section.
- Page 8 line 255; 30 % solution (should be w/v).
- The term “Discussion” is included in the footnote of Figure 4.
- The references include the same ordinal number twice.
- Reference 26 in incomplete.
Author Response
See attached PDF

Reviewer 2 Report
The study by Unrau et all. investigates the effect of myeloid Smad7 Deficiency in CCl4 liver injury.
Major Concerns:
- There is high variation between the mice (independent of treatment) – data don’t seem to follow Gaussian distributon, but rather cluster in two groups – ANOVA seems to be impermissible.
- Why do not all mice respond to the treatment? Is this caused by merging the 5 independent experiments?
- Smad7 knockout should be shown in different myeloid cells? Are Kupffer cells also affected?
- What is the effect of Smad7 knockout on TGF-b signaling in myeloid cells? Is there a redundancy between Smad6 and Smad7? If yes, please provide at least in vitro data with Smad6+Smad7 deficiency.
- How is the highly significant decrease in neutrophiles in smad7-KO mice explained?
- In Fig.3 “not determined” levels seem to be used for statistics.
- TGF-b signaling should be analyzed in the livers, but also in isolated hepatocytes and Kupffer cells
- Is there a change in body weight/liver weight?
- The authors apply the same model they published in 2019 (48 h treatment vs. 6 week treatment, almost identical read-outs)– no new insights are generated.
Author Response
see attached PDF

Round 2
Reviewer 2 Report
The authors addressed all major concerns.